# Near-extremal limits of warped CFTs

**Ankit Aggarwal[1,2]\*, Alejandra Castro[3]†, Stéphane Detournay[1]‡ and Beatrix Mühlmann[4]°**

**1** Physique Mathematique des Interactions Fondamentales, Universite Libre de Bruxelles, Campus Plaine - CP 231, 1050 Bruxelles, Belgium

**2** Institute for Theoretical Physics Amsterdam and Delta Institute for Theoretical Physics, University of Amsterdam, Science Park 904, 1098 XH Amsterdam, The Netherlands

**3** Department of Applied Mathematics and Theoretical Physics, University of Cambridge, Cambridge CB3 0WA, United Kingdom

**4** Department of Physics, McGill University, Montreal QC H3A 2T8, Canada

\* ankit.aggarwal@ulb.be , † ac2553@cam.ac.uk ,
‡ sdetourn@ulb.ac.be , ° beatrix.muehlmann@mcgill.ca

## Abstract

Warped conformal field theories (WCFTs) are two-dimensional non-relativistic systems, with a chiral scaling and shift symmetry. We present a detailed derivation of the near-extremal limit for their torus partition function. This limit requires large values of the central charge, and is only consistent for non-unitary WCFT. We compare our analysis with previous studies of WCFT and its relation to a one-dimensional warped-Schwarzian theory. We discuss different ensembles of warped CFTs and contrast our results with analogous limits in two-dimensional CFTs.



# 1 Introduction

Warped Conformal Field Theories (WCFTs) are rather peculiar two-dimensional systems. They were conceived in a holographic era, where their properties and utility were defined by their ties to gravitational systems [1–4]. But they suffer from an identity crisis: they mimic some aspects of two-dimensional CFTs, making them at times a masquerader. This is problematic since, in a holographic setup, it creates ambiguities on how to identify a bulk and boundary pair.

In this work, we aim to give WCFTs their own identity, and further attempt to distinguish them from their CFT$_2$ cousins. The direction we will pursue is to investigate the so-called near-extremal limit. This is motivated by the universal behaviour decoded in [5]: CFT$_2$ that admit a large central charge limit will generically contain a sector described a Schwarzian theory. The limit was inspired by the behaviour of black holes near-extremality, where the temperature is low and the angular momentum is large. Still, the setup and outcomes do not require a holographic dual. One can show that the near-extremal limit gives access to an interesting regime of the CFT and probes it without requiring detailed information. That is, it is a universal sector of the theory.

At the level of the torus partition function, we will show under which circumstances a WCFT also has an interesting near-extremal limit. Our reason to focus on this quantity is motivated by the masquerade. In particular, at high temperatures WCFTs mimic the Cardy regime of a CFT$_2$ [3]. Here we will explore the opposite regime, and determine if there are significant differences in the setup of the limit and outcomes of it. As we will see, the near-extremal limit of a WCFT will again share many properties that are also present in a CFT$_2$. For example, in a fixed angular momentum ensemble, the resulting answer in both systems is naively identical. But there are also some important differences. We will elaborate on these differences and similarities in detail in Sec. 4.

Relating WCFTs to a one-dimensional Schwarzian theory has been done in the past. In particular, the work of [6,7] connects the symmetries of a WCFT to those of a warped Schwarzian theory. The basic properties of this one-dimensional system are as follows. It contains the usual Schwarzian sector, where there is one degree of freedom $f(u)$ that describes the reparametrization of a circle.[1] The warped Schwarzian also has an internal $\hat{u}(1)$ symmetry; these come about naturally when studying the complex SYK system [13].[2] The basic structure of the Euclidean action is

$$S_{\text{w-schw.}} = K \int_0^{\tilde{\beta}} \mathrm{d}u \, g'(u)^2 - C \int_0^{\tilde{\beta}} \mathrm{d}u \left\{ \tan\left( \frac{\pi f(u)}{\tilde{\beta}} \right), u \right\}. \tag{1}$$

Here $u$ is the coordinate on the circle, whose period $\tilde{\beta}$ defines the inverse temperature, $g(u)$ is the mode associated to the $\hat{u}(1)$, and the last term is a Schwarzian derivative. In the notation of the complex SYK system, $K$ is related to the compressibility and $C$ to the specific heat. The work of [6,7] show how to relate these quantities to data of the WCFT, and illustrate how correlation functions and the path integral are related.

In this work, we add to these relations by carefully defining the near-extremal limit of a WCFT. This is done by demanding that the torus partition function has a universal behaviour, where the vacuum character dominates. We will show that it is only possible to access a sector that resembles the warped Schwarzian theory if the theory is non-unitary, among other restrictions on the vacuum state. In a WCFT, it is also interesting to study non-local transformation of the algebra — this is how the theory pretends to be a CFT$_2$. We will consider two cases. The

---

[1]This theory has been prominent in recent years due to its connection to 2D quantum gravity and the SYK model [8–11]. The path integral of the Schwarzian theory was elegantly discussed in [12].

[2]The warped Schwarzian theory also appears through a Hamiltonian reduction of Lower-Spin gravity [14], the minimal holographic set-up for WCFTs consisting of an $SL(2,\mathbb{R}) \times U(1)$ Chern-Simons theory [15].

first case is when the algebra is in its canonical (local) form, this is usually called the "canonical ensemble"; the second case corresponds to a quadratic transformation that introduces state dependence in the symmetry algebra, which is known as the "quadratic ensemble". We will see how the responses in the near-extremal limit differ in these two ensembles, and discuss its consequences.

Although these new improvements are technical, and might seem overscrupulous, we believe that it is important to highlight differences and limitations of the limit when making connections with a dual gravitational system, and even the complex SYK system. In a companion work [16], we will discuss the holographic counterpart of our results in depth. We will use warped AdS$_3$ black holes (see [17] and references therein) as solutions to topologically massive gravity, and carefully setup the holographic dictionary to see how the different facets of the near-extremal limit of WCFTs make an appearance in the near-extremal limits of the warped black hole.

This paper is organised as follows. In Sec. 2, we review the spectrum and torus partition function of WCFT in the canonical ensemble. We construct and analyse the near-extremal limit by also going to fixed angular momentum and fixed energy ensembles. In Sec. 3, we translate the WCFT to the quadratic ensemble, construct its near-extremal limit and repeat the near-extremal analysis of the canonical ensemble. We end in Sec. 4 with a comparison of the different ensembles (canonical and quadratic) and also compare to the near-extremal limit of a CFT$_2$. The two appendices compare the near-extremal limit to the Cardy limit (see App. A) and review the near-extremal limit of a CFT$_2$ (see App. B).

## 2 Canonical ensemble

A Warped Conformal Field Theory (WCFT) is a two-dimensional non-relativistic quantum field theory, and the defining property is its symmetries: provided two coordinates $(\varphi, t)$, the system is invariant under [2,3]

$$\varphi \;\rightarrow\; f(\varphi), \qquad t \;\rightarrow\; t + g(\varphi). \tag{2}$$

Here $f(\varphi)$ is a diffeomorphism and $g(\varphi)$ an arbitrary function. Their effects are to introduce a scaling symmetry for $\varphi$, while $t$ only has a shift symmetry which makes the theory non-relativisitic. These local symmetries are described by a Virasoro-Kac-Moody algebra, and the global subgroup is an $sl(2, \mathbb{R}) \times \hat{u}(1)$ algebra.[3] In this sense, a WCFT is reminiscent of a CFT$_2$, but still distinct with its own attributes.

It is useful to record some basic properties of the algebra and generators of a WCFT. We denote by $L_n$ the Virasoro generators and $P_n$ the $\hat{u}(1)$ Kac-Moody generators, with $n \in \mathbb{Z}$. The commutation relations are

$$
\begin{aligned}
[L_n, L_{n'}] &= (n - n') L_{n+n'} + \frac{c}{12} n(n^2 - 1) \delta_{n,-n'}, \\
[L_n, P_{n'}] &= -n' P_{n'+n}, \\
[P_n, P_{n'}] &= \kappa \frac{n}{2} \delta_{n,-n'},
\end{aligned}
\tag{3}
$$

where $c$ is the central charge and $\kappa$ is the $\hat{u}(1)$ level. In our choice of coordinates, and their transformations (2), we have selected $\varphi$ as an angular variable with period $\varphi \sim \varphi + 2\pi$, and $t$ as a time variable in Lorentzian signature.[4] In this context, we identify $L_0$ with the angular

---

[3]Notice that a distinct global subgroup exists, consisting in the centrally extended two-dimensional Poincaré group [18].

[4]This choice is motivated by how the transformations (2) make an appearance as the asymptotic symmetry group of gravitational systems. See, for example, [3,4].

momentum of the system, and $P_0$ with the Hamiltonian. Further properties and aspects of this algebra can be found in [3, 19, 20].

Our main focus in this section is to analyse the torus partition function of a WCFT, and show that it has an elegant universal behaviour at low temperatures. This is analogous to the near-extremal behaviour of the torus partition function for CFT$_2$ [5], which we review in App. B. With the purpose of building and decoding the near-extremal limit for WCFTs, we will start this section with an overview of the spectrum of WCFTs and the modular properties of its partition function; in the later part, we propose a definition of the near-extremal limit for WCFT, discuss when the limit is interesting, and display its universal features.

## 2.1 Spectrum and torus partition function

In this subsection, we revise various properties of the spectrum and the torus partition function. This is based on the results presented in [19, 20], which we refer to for further details.

We start by simply placing our WCFT on a torus. This means that our two coordinates will have identifications

$$(t, \varphi) \sim (t, \varphi + 2\pi) \sim (t + 2\pi z, \varphi + 2\pi \tau). \tag{4}$$

Here we have taken the canonical torus, where the spatial identification has unit radius. For the thermal identification, we are using complex variables $(\tau, z)$, which are related to the angular potential $\vartheta$ and the inverse temperature $\beta$ via

$$\tau \equiv \frac{i\vartheta}{2\pi}, \qquad z \equiv \frac{i\beta}{2\pi}. \tag{5}$$

The modular group is described by two transformations: $S$ which corresponds to interchanging the spatial and thermal cycles; and $T$ which accounts for adding the spatial cycle to the thermal cycle.

The torus partition function is given by

$$Z(\tau, z) = \text{Tr}\left(q^{L_0} y^{P_0}\right), \quad q \equiv e^{2\pi i \tau}, \quad y \equiv e^{2\pi i z}, \tag{6}$$

where the generators $L_0$ and $P_0$ are defined on the cylinder. In conventions where the conjugate potentials are purely real, we will have $\tau$ and $z$ purely imaginary. In relation to the notation in [19], this would be the partition function $\hat{Z}(z|\tau)$; as shown there under a modular $S$-transformation one finds

$$Z(\tau, z) = e^{-i\pi\kappa \frac{z^2}{2\tau}} Z\left(-\frac{1}{\tau}, \frac{z}{\tau}\right), \tag{7}$$

where $\kappa$ is the $\hat{u}(1)$-level. That is, the theory has an anomaly under an $S$-transformation. In the following subsections, we will investigate properties of (6). An important portion of that analysis relies on how the torus partition function is organized in terms of its spectrum, i.e., its primary and descendant states, which we turn to now.

The most straightforward way to construct and specify the spectrum of the system is by demanding that representations of the symmetry algebra fall into unitary representations. This can be implemented for a WCFT, and it will impose restrictions on the central extensions $(c, \kappa)$ and the eigenvalues of the zero-mode charges $(L_0, P_0)$. However, due to the holographic properties of WCFTs, it becomes natural to loosen these restrictions and allow for some violations of unitarity while still complying with interesting and well-defined observables in the system. For this reason, we will also allow for the $\hat{u}(1)$-level to be negative.

We will organize the spectrum in terms of primaries and descendants of the Virasoro-Kac-Moody algebra (3). For primary states, we will denote by $h - c/24$ and $p$ the eigenvalues with respect to $L_0$ and $P_0$ on the cylinder, respectively. The definition of a primary is a state $|h, p\rangle$ that obeys

$$L_n |h, p\rangle = P_n |h, p\rangle = 0, \qquad n > 0. \tag{8}$$

Descendants are created by acting with $L_{-n}$ and $P_{-n}$ ($n > 0$) arbitrarily many times. The vacuum state will have $(h_{\text{vac}}, p_{\text{vac}})$, and it is defined as the state annihilated also by $L_{-1}$, in addition to (8).[5] In the following, we will describe different irreducible representations of the Virasoro-Kac-Moody algebra as done in [20], which includes unitary representations and certain classes of non-unitary ones. In this context, there are a few important assumptions made in the analysis:

1. We will always take $c \geq 2$.
2. Primary states will always have semi-positive definite norms.
3. Primary states satisfy

$$h \geq \begin{cases} p^2/\kappa, & \text{if } p^2 < 0, \\ 0, & \text{if } p^2 \geq 0. \end{cases} \tag{9}$$

The three cases below report on the characters $\chi_{h,p}(\tau, z)$ that encode the descendants for the vacuum state and general primary states, and the detailed derivation is in [20]. With this, we will organize the partition function (6) in terms of these characters.

**Case 1 (non-unitary): $\kappa < 0$ and $p$ real.** This is a representation that violates unitarity, still the character is well defined. Adding up the contribution over all descendants of a primary with conformal weight $h$ and real charge $p$, the character is

$$\chi_{h,p}^{(1)}(\tau, z) = \frac{1}{\eta(2\tau)} q^{h - \frac{c-2}{24}} y^p (1 - \delta_{\text{vac}} q), \qquad p \in \mathbb{R}. \tag{10}$$

Here, $\eta(\tau)$ is the Dedekind eta function. The variable $\delta_{\text{vac}}$ has support only on the vacuum, i.e., $\delta_{\text{vac}} = 1$ for the vacuum state only. This has the effect of removing the null state created by acting with $L_{-1}$, as it is standard for a $\text{CFT}_2$. The hermiticity condition on the states is $P_n^\dagger = P_{-n}$ and $L_n^\dagger = L_{-n}$; this implies that some descendants have negative norm states since $\kappa$ is negative. Still all the coefficients in (10) are positive.

**Case 2 (non-unitary): $\kappa < 0$ and $p$ imaginary.** Our second scenario considers primaries for which their charge is purely imaginary. In this case the hermiticity condition is $P_n^\dagger = -P_{-n}$, and $L_n^\dagger = L_{-n}$. In this case all states make a positive contribution to the character. The corresponding result is

$$\chi_{h,p}^{(2)}(\tau, z) = \frac{1}{\eta(\tau)^2} q^{h - \frac{c-2}{24}} y^p (1 - \delta_{\text{vac}} q), \qquad p \in i\mathbb{R}. \tag{11}$$

One of the interesting results in [20] was to show that any WCFT with $\kappa < 0$ must feature at least two primary states with imaginary $\hat{u}(1)$ charge, which comes from imposing crossing symmetry of the torus partition function. We will use this in what follows.

**Case 3 (unitary): $\kappa > 0$ and $p$ real.** The final case is the more familiar context that relates to $\text{CFT}_2$, where the level is positive. Here the character reads

$$\chi_{h,p}^{(3)}(\tau, z) = \frac{1}{\eta(\tau)^2} q^{h - \frac{c-2}{24}} y^p (1 - \delta_{\text{vac}} q), \qquad p \in \mathbb{R}. \tag{12}$$

The hermiticity condition used is the usual one: $P_n^\dagger = P_{-n}$ and $L_n^\dagger = L_{-n}$, hence states have positive norm since $\kappa$ is positive.

---

[5]In contrast to $\text{CFT}_2$ the conformal weights and charge of the vacuum state are not fixed by the symmetries. The only condition imposed by demanding invariance under $sl(2, \mathbb{R}) \times u(1)$ is that $h_{\text{vac}} = p_{\text{vac}}^2/\kappa$ [3].

Having gathered the characters that are relevant for our discussion, we will decompose the torus partition function as follows. For a **unitary** WCFT, i.e., $\kappa > 0$, we will take

$$Z(\tau, z)_{\mathrm{u}} = \sum_{\substack{\text{primaries} \\ +\text{vacuum}}} \chi^{(3)}_{h,p}(\tau, z). \tag{13}$$

Here, all states satisfy $h \geq p^2/\kappa$ and $p \in \mathbb{R}$; the sum over primaries also includes the contribution of the vacuum state. For a **non-unitary** WCFT, i.e., $\kappa < 0$, we will take

$$Z(\tau, z)_{\cancel{\mathrm{u}}} = \sum_{\substack{\text{primaries} \\ p \in \mathbb{R}}} \chi^{(1)}_{h,p}(\tau, z) + \sum_{\substack{\text{primaries} \\ p \in i\mathbb{R}}} \chi^{(2)}_{h,p}(\tau, z). \tag{14}$$

This partition function includes as well a contribution from the vacuum state; for now we will be agnostic if it is a state with real or imaginary $p_{\mathrm{vac}}$ since symmetries alone do not fix the properties of the state. As we will see in the next subsection, the requirement of a well-defined near-extremal limit will impose restrictions on $p_{\mathrm{vac}}$.

## 2.2 Near-extremal limit

To construct a near-extremal limit, we will implement a procedure similar to that in a CFT$_2$ [5],[6] while being adapted to the symmetries and characteristics of WCFTs. This will bring some similarities, but also contrasts among a CFT$_2$ and a WCFT. In this context, there are two important aspects that are key to emphasise:

1. The near-extremal limit is *not* a limit within the usual Cardy-regime. Their regimes of validity do not overlap.
2. The essence of a well-defined and interesting near-extremal limit is the dominance of the vacuum contribution.

In the following, we will discuss how one can construct and decipher the near-extremal limit for the two classes of torus partition functions.

**Unitary WCFTs.** We start with the unitary cases with the aim to illustrate the subtleties and obstructions that arise. For this case, it will suffice to define near-extremality as a regime where

$$\tau \to i\infty, \qquad z \to i\infty, \tag{15}$$

and for simplicity we will choose $z/\tau$ fixed. Note that sending $\tau \to i\infty$ will be the natural choice to project onto the vacuum, and our conclusions will not change if $z$ is fixed or small. The torus partition function is given by (13). Using the modular $S$-transform we can write

$$
\begin{aligned}
Z(\tau, z)_{\mathrm{u}} &= e^{-i\pi\kappa \frac{z^2}{2\tau}} Z\left(-\frac{1}{\tau}, \frac{z}{\tau}\right)_{\mathrm{u}} \\
&= e^{-i\pi\kappa \frac{z^2}{2\tau}} \chi^{(3)}_{\mathrm{vac}}\left(-\frac{1}{\tau}, \frac{z}{\tau}\right) + e^{-i\pi\kappa \frac{z^2}{2\tau}} \sum_{\text{primaries}} \chi^{(3)}_{h,p}\left(-\frac{1}{\tau}, \frac{z}{\tau}\right).
\end{aligned}
\tag{16}
$$

Taking a ratio, we find

$$\frac{Z(\tau, z)_{\mathrm{u}}}{e^{-i\pi\kappa \frac{z^2}{2\tau}} \chi^{(3)}_{\mathrm{vac}}\left(-\frac{1}{\tau}, \frac{z}{\tau}\right)} = 1 + \frac{1}{1 - e^{-2\pi i \frac{1}{\tau}}} \sum_{\text{primaries}} e^{-2\pi i \frac{1}{\tau}(h - h_{\mathrm{vac}})} e^{2\pi i \frac{z}{\tau}(p - p_{\mathrm{vac}})}. \tag{17}$$

---

[6]See App. B for a review.

In the limit (15), we see that the prefactor in the sum diverges polynomially with $\tau$. Unfortunately, the sum does not suppress this prefactor: the exponential terms that depend on the conformal weights go to one; since $p \in \mathbb{R}$ the sum over $U(1)$-charges is oscillatory. Note that this cannot be fixed by adjusting $z$ in (15): the main issue is that $z/\tau$ is real.

We therefore conclude that unitary WCFTs do not have a near-extremal limit where the vacuum character dominates.

**Non-unitary WCFTs.** This case is interesting and will lead to a desirable outcome. We start by defining the near-extremal limit as

$$\tau \to i\infty, \qquad \Omega \equiv \frac{\tau}{z} \ll 1. \tag{18}$$

Here, we have introduced the angular velocity $\Omega$, which in terms of the real potentials is given by $\vartheta = \beta\Omega$. Along the lines of [5], this limit will be refined and placed in the context of holographic theories as we deconstruct its consequences and validity.

Our main aim with this limit is to establish within the regime (18) when the vacuum character will dominate the torus partition function. For a non-unitary WCFT we have to specify in addition which state is the vacuum state: recall that the global symmetries of the theory only fix $h_{\text{vac}} = p_{\text{vac}}^2/\kappa$, but nothing else. If $p_{\text{vac}}$ is real, we will end up with the same conclusion as in the unitary case described above. However, if $p_{\text{vac}}$ is purely imaginary our problems can be circumvented! Assuming that the vacuum state has a purely imaginary value we find

$$\frac{Z(\tau,z)_{\not\vphantom{l}}}{e^{-i\pi\kappa\frac{z^2}{2\tau}}\chi_{\text{vac}}^{(2)}\left(-\frac{1}{\tau},\frac{z}{\tau}\right)} = 1 + \frac{1}{1 - e^{-2\pi i\frac{1}{\tau}}} \sum_{\substack{\text{primaries} \\ p\in i\mathbb{R}}} e^{-2\pi i\frac{1}{\tau}(h-h_{\text{vac}})} e^{\frac{2\pi}{\Omega}i(p-p_{\text{vac}})} + \dots \tag{19}$$

Here, we have used the modular transformation (7) on (14). The dots are the contributions for the primaries with real values of $p$ which we address momentarily. In the limit (18), we see that again we have a growing polynomial contribution in $\tau$; in order to suppress this divergence, we need a damping from the sum over primaries with imaginary $p$ in (19). This can be achieved by demanding that $p_{\text{vac}}$ be the state that bounds the imaginary charge, i.e.,

$$ip_{\text{vac}} > ip, \qquad \forall\, p \in i\mathbb{R}. \tag{20}$$

To suppress the terms in $Z(\tau,z)_{\not\vphantom{l}}$ with $p \in \mathbb{R}$, we then need $ip_{\text{vac}} > 0$. With this, we can suppress the contributions from all primaries in (19), and therefore, in the limit (18), it is a good approximation to write

$$Z(\tau,z)_{\not\vphantom{l}} \approx e^{-i\pi\kappa\frac{z^2}{2\tau}}\chi_{\text{vac}}^{(2)}\left(-\frac{1}{\tau},\frac{z}{\tau}\right). \tag{21}$$

We can further cast (21) by taking the near-extremal limit on the vacuum characters

$$\begin{aligned}
\chi_{\text{vac}}^{(2)}\left(-\frac{1}{\tau},\frac{z}{\tau}\right) &\approx -\frac{2\pi}{\tau^2}e^{-\frac{\pi i}{6}\tau}e^{-2\pi i\frac{1}{\tau}(h_{\text{vac}}-\frac{c-2}{24})}e^{2\pi\frac{z}{\tau}ip_{\text{vac}}} \\
&\approx 2\pi\left(\frac{2\pi}{\vartheta}\right)^2 e^{\frac{\vartheta}{12}-\frac{4\pi^2}{\vartheta}(h_{\text{vac}}-\frac{c-2}{24})}e^{2\pi\frac{\beta}{\vartheta}ip_{\text{vac}}}.
\end{aligned} \tag{22}$$

In the first line, we used (11) for the vacuum state, and the properties of the Dedekind eta function (B.12); in the second line, the expression is re-written in terms of the real potentials $\vartheta$ and $\beta$. Notice that we are being somewhat careless here regarding which terms we are

keeping, and hence we should revisit (18). We would like to keep contributions related to the vacuum state, in particular $h_{\text{vac}}$, and the central charge $c$. For this reason, we will take

$$c \gg 1, \tag{23}$$

and $h_{\text{vac}}$ to depend on $c$.[7] For the potentials we will take

$$\vartheta \sim c, \qquad \beta \sim c^{\alpha}, \tag{24}$$

with $\alpha > 1$, such that $\Omega \ll 1$ in the large central charge limit. With this, we can then write (21) as

$$Z(\vartheta, \beta)_{\not M} \approx 2\pi \left( \frac{2\pi}{\vartheta} \right)^2 e^{\frac{\beta^2}{\vartheta} \frac{\kappa}{4} + \frac{\vartheta}{12} - \frac{4\pi^2}{\vartheta} \left( h_{\text{vac}} - \frac{c}{24} \right) + 2\pi \frac{\beta}{\vartheta} i p_{\text{vac}}}, \tag{25}$$

which is the universal behaviour at low temperatures of a non-unitary WCFT. Note that we have implemented a large-$c$ limit in (25). At this stage one can compare with the partition function of a warped Schwarzian theory (1) as done in [6,7]. We find good agreement between (25) and the 1D theory. In particular, the polynomial dependence in $\vartheta$ agrees with the counting of zero modes: four of them coming from $sl(2) \times u(1)$, each contributing with a power of $\vartheta^{-1/2}$. Also, the quadratic dependence in $\beta$ inside the exponent is directly related to the quadratic term in (1): this nicely connects the anomaly in the $S$-transformation of (7) to the $\hat{u}(1)$ symmetry in 1D.

In the following, we will extract the entropy and other information from this expression by going to the appropriate ensembles. Following the conventions we had at the start of the section, we will define

$$E \equiv \langle P_0 \rangle, \qquad J \equiv \langle L_0 \rangle, \tag{26}$$

i.e., the energy and angular momentum of the system as the expectation values of the zero modes of the algebra.

**Fixed $(\vartheta, E)$ ensemble.** For fixed $E$, we consider the Laplace transform of (25)

$$Z_E(\vartheta) = \int_0^\infty \mathrm{d}\beta \, e^{\beta E} Z(\vartheta, \beta)_{\not M}. \tag{27}$$

This integral can be easily approximated by a saddle point approximation; the location of the saddle point is at

$$\beta_* = -\frac{2}{\kappa} (E\vartheta + 2\pi i p_{\text{vac}}). \tag{28}$$

Since $\kappa < 0$, and $p_{\text{vac}}$ is purely imaginary, this saddle is indeed in the range of (27). For this saddle to be within the near-extremal limit (24), we also require that $E \sim c^{\alpha-1}$, which implies that $E$ is very large. In the saddle point approximation we obtain

$$Z_E(\vartheta) \approx 96\pi^2 \left( -\frac{6}{c^3 \kappa} \right)^{1/2} \exp\left( -4\pi \frac{i p_{\text{vac}}}{\kappa} E \right) \exp\left( -\frac{E^2}{\kappa} \vartheta \right) \left( \frac{c\pi}{6\vartheta} \right)^{3/2} \exp\left( \frac{\pi^2 c}{6\vartheta} \right). \tag{29}$$

It is interesting that $h_{\text{vac}}$ dropped out of this expression; it is because it appears as $h_{\text{vac}} - p_{\text{vac}}^2/\kappa$ which is zero. Here we have grouped the terms depending on how $\vartheta$ enters in the expressions. We will interpret it as

$$Z_E(\vartheta) = e^{S_0 - \vartheta J_0} Z_{\text{w-schw.}} \left( \frac{6\vartheta}{c} \right). \tag{30}$$

---

[7]In many holographic settings $h_{\text{vac}}$ is independent of the central charge, so it could be dropped for that reason. It will be instructive to keep it, and hence assume it is a large number.

The first contribution, $S_0$, is identified with the extremal entropy. In other words, it captures terms that do not depend on $\vartheta$:

$$S_0 = -4\pi \frac{ip_{\text{vac}}}{\kappa} E - \frac{1}{2} \log\left((-\kappa)c^3\right) + \dots, \tag{31}$$

where the dots are subleading terms as $c \gg 1$. As it has been discussed in [5, 21], $S_0$ should not be interpreted as a ground state entropy since the near-extremal limit makes the theory gapless. The next term in (30) is the "extremal" angular momentum, which in this system reads

$$J_0 = \frac{E^2}{\kappa} + \dots \tag{32}$$

Note that the leading contribution to $J_0$ is negative; this is still within our unitary bounds (9). Finally, we have the last contribution which reads

$$Z_{\text{w-schw.}}\left(\frac{6\vartheta}{c}\right) = \left(\frac{c\pi}{6\vartheta}\right)^{3/2} \exp\left(\frac{\pi^2 c}{6\vartheta}\right). \tag{33}$$

This expression is the same as one obtains in a CFT$_2$; see (B.23). Therefore, it is tempting to interpret it as a Schwarzian effective action controlling the low-temperature dynamics due to the polynomial behaviour of $\vartheta^{-3/2}$. However, one should be cautious with this interpretation here. As explained below (25) the more natural interpretation is in terms of the warped Schwarzian theory, where the polynomial correction is quadratic. The change from $\vartheta^{-2}$ to $\vartheta^{-3/2}$ is due to the anomaly in the S-transform (7) as we perform the Legendre transform. (The second derivative of this anomaly introduces an extra power of $\vartheta$ as one does the saddle point approximation of (27).) The expression quoted in (33) is the partition function in an ensemble where the $\hat{u}(1)$-charge, $\langle P_0 \rangle$, is fixed in the warped Schwarzian theory; it just happens to coincide with the Schwarzian answer.

**Fixed $(\beta, J)$ ensemble.**   The fixed $J$ ensemble can be obtained in a similar fashion as above. We have

$$Z_J(\beta) = \int \mathrm{d}\vartheta\, e^{\vartheta J} Z(\vartheta, \beta)_{\not{u}}, \tag{34}$$

with $Z(\vartheta, \beta)_{\not{u}}$ given in (25). The integrand is extremized for

$$\vartheta_*^2 = \frac{1}{J + \frac{1}{12}}\left(\frac{\kappa}{4}\beta^2 + 2\pi\beta ip_{\text{vac}} - 4\pi^2\left(h_{\text{vac}} - \frac{c}{24}\right)\right). \tag{35}$$

As before we need to assure that the location of the saddle point lays within (24). In this case, this requires that $J$ is negative and scales like $|J| \sim c^{2(\alpha-1)}$. On the other hand, we have a unitary bound (9), where $J = h - \frac{c}{24}$ with $h \geq 0$. Hence, to have a consistent limit and approximation, we require that $1 < \alpha \leq \frac{3}{2}$.

Taking these aspects into account, and using the positive root in (35), the result of the saddle point approximation is

$$Z_J(\beta) \approx 48\pi^2\left(\frac{6}{|J|}\right)^{1/2} \exp\left(4\pi ip_{\text{vac}}\sqrt{\frac{J}{\kappa}} + \sqrt{\kappa J}\beta\right)\left(\frac{c\pi}{3\beta}\sqrt{\frac{J}{\kappa}}\right)^{3/2} \exp\left(\frac{\pi^2 c}{3\beta}\sqrt{\frac{J}{\kappa}}\right). \tag{36}$$

Casting this expression in terms of

$$Z_J(\beta) = e^{S_0 - \beta E_0} Z_{\text{w-schw.}}(\tilde{\beta}), \tag{37}$$

in the large $|J|$ and $c$ limit we have

$$E_0 = -\sqrt{\kappa J} + \dots,$$
$$S_0 = 4\pi i p_{\text{vac}} \sqrt{\frac{J}{\kappa}} - \frac{1}{2}\log(|J|) + \dots, \tag{38}$$
$$\tilde{\beta} = \frac{3}{c}\sqrt{\frac{J}{\kappa}}\beta,$$

and $Z_{\text{w-schw.}}(\tilde{\beta})$ is defined in (33). Note that this is the only case where the effective temperature $\tilde{\beta}$ depends on the extremal parameter $J$ and also the level $\kappa$. For all other cases considered here, only the central charge and numerical factors enter in the temperature dependence of the leading correction to the partition function.

The anomaly in the $S$-transform (7) is again the culprit in changing the scaling of the partition function from $\vartheta^{-2}$ to $\vartheta^{-3/2}$. In this case the anomaly controls the location of the saddle point and it also affects the scaling of the integrand.

## 3 Quadratic ensemble

WCFTs are infamous because they can disguise themselves as conformal field theories under certain circumstances. Basically, one can perform a non-local reparametrization of the theory that restores modular invariance in the partition function, at the cost of making the Virasoro-Kac-Moody algebra non-local. For this reason, in this section we will explore the near-extremal limit under these circumstances, and contrast our findings with a CFT$_2$ and a WCFT in its canonical form.

### 3.1 Spectrum and torus partition function

As observed in [3], it is interesting to consider the following redefinition of the Virasoro-Kac-Moody generators in Sec. 2,[8]

$$\mathcal{L}_n = L_n - \frac{2}{\kappa}P_0 P_n + \frac{1}{\kappa}P_0^2 \delta_n, \qquad \mathcal{P}_n = -\frac{2}{\kappa}P_0 P_n + \frac{1}{\kappa}P_0^2 \delta_n. \tag{39}$$

This is a quadratic transformation on the generators, and hence the name "Quadratic ensemble" (see [4,22] for a bulk realization of quadratic ensemble). The upshot of this transformation is that the algebra (3) keeps its structure,

$$[\mathcal{L}_n, \mathcal{L}_m] = (n-m)\mathcal{L}_{n+m} + \frac{c}{12}n^3 \delta_{n+m},$$
$$[\mathcal{L}_n, \mathcal{P}_m] = -m\mathcal{P}_{n+m} + m\mathcal{P}_0 \delta_{n+m}, \tag{40}$$
$$[\mathcal{P}_n, \mathcal{P}_m] = -2n\mathcal{P}_0 \delta_{n+m},$$

but now with the feature that the zero mode $\mathcal{P}_0$ plays the role of the $\hat{u}(1)$-level. This is one of the ways in which the algebra is non-local, since it is state dependent.

Our aim is to describe properties of the torus partition function in this ensemble, so we start by collecting some facts about the spectrum and characters. The construction of the quadratic ensemble spectrum relies on properties inherited from the canonical ensemble. To exploit this, one starts with the relation of the zero-mode generators, which reads

$$\mathcal{L}_0 = L_0 - \frac{P_0^2}{\kappa}, \qquad \mathcal{P}_0 = -\frac{P_0^2}{\kappa}. \tag{41}$$

---

[8]Note that we are following the conventions in [20], which differ from [3].

The definitions of a primary state and the descendants, in the quadratic ensemble are taken to be the same as those stated around (8). It is useful to note that by using (41), the primary states in the canonical ensemble $|h, p\rangle$ are also the primary states in the quadratic ensemble with weights

$$
\begin{aligned}
\langle \mathcal{L}_0 \rangle &= h - \frac{p^2}{\kappa} - \frac{c}{24}, \\
\langle \mathcal{P}_0 \rangle &= -\frac{p^2}{\kappa}.
\end{aligned}
\tag{42}
$$

Note that two states $|h, p\rangle$ and $|h, -p\rangle$ have the same eigenvalues with respect to $\mathcal{L}_0$ and $\mathcal{P}_0$. Therefore, to avoid any confusion due to such degeneracies, we will continue to label the states by $(h, p)$. In this way, the operator content in the canonical and quadratic ensemble remains the same. We also see that the vacuum eigenvalues in the quadratic ensemble are

$$
\langle \mathcal{L}_0 \rangle_{\text{vac}} = -\frac{c}{24}, \qquad \langle \mathcal{P}_0 \rangle_{\text{vac}} = -\frac{1}{\kappa} p_{\text{vac}}^2.
\tag{43}
$$

Here, we used that $h_{\text{vac}} = p_{\text{vac}}^2 / \kappa$. One appealing outcome of this ensemble is that $\langle \mathcal{L}_0 \rangle_{\text{vac}}$ is fixed by the central charge. Also, these two quantities could be confused for a left and a right Virasoro central charges, while the symmetries only consist of algebra (40).

The grand canonical partition function in the quadratic ensemble is defined as

$$
Z(\beta_L, \beta_R) = \text{Tr}\, e^{-\beta_R \mathcal{P}_0 - \beta_L \mathcal{L}_0},
\tag{44}
$$

where $\beta_L$ and $\beta_R$ are real potentials. What was shown in [3], using the transformation properties of the $\mathcal{L}_0$ and $\mathcal{P}_0$ inherited from the canonical ensemble, is that

$$
Z(\beta_L, \beta_R) = Z\left(\frac{4\pi^2}{\beta_L}, \frac{4\pi^2}{\beta_R}\right),
\tag{45}
$$

i.e., the partition function is modular invariant. The anomaly appearing in (7) is no longer present, and this will be important in our next subsection.

The characters in the quadratic ensemble can be found in the same way as for the canonical ensemble except that the weights of the primary states are now given by (42). Following the same notation as in Sec. 2.1, we have

$$
\textbf{Case 1}: \quad \kappa < 0,\ p \in \mathbb{R}, \quad \chi_{h,p}^{(1)}(\beta_L, \beta_R) = \frac{1}{\eta\left(\frac{i\beta_L}{\pi}\right)} q_L^{h - p^2/\kappa - (c-2)/24} q_R^{-p^2/\kappa}(1 - \delta_{\text{vac}} q_L),
$$

$$
\textbf{Case 2}: \quad \kappa < 0,\ p \in i\mathbb{R}, \quad \chi_{h,p}^{(2)}(\beta_L, \beta_R) = \frac{1}{\eta\left(\frac{i\beta_L}{2\pi}\right)^2} q_L^{h - p^2/\kappa - (c-2)/24} q_R^{-p^2/\kappa}(1 - \delta_{\text{vac}} q_L), \tag{46}
$$

$$
\textbf{Case 3}: \quad \kappa > 0,\ p \in \mathbb{R}, \quad \chi_{h,p}^{(3)}(\beta_L, \beta_R) = \frac{1}{\eta\left(\frac{i\beta_L}{2\pi}\right)^2} q_L^{h - p^2/\kappa - (c-2)/24} q_R^{-p^2/\kappa}(1 - \delta_{\text{vac}} q_L),
$$

where we defined $q_R \equiv e^{-\beta_R}$ and $q_L \equiv e^{-\beta_L}$. Organising the partition function as a sum over characters, we find

$$
\begin{aligned}
\textbf{Unitary}: \quad Z(\beta_L, \beta_R)_{\text{u}} &= \sum_{\substack{\text{primaries} \\ +\text{vacuum}}} \chi_{h,p}^{(3)}(\beta_L, \beta_R), \\
\textbf{Non-unitary}: \quad Z(\beta_L, \beta_R)_{\text{\not u}} &= \sum_{\substack{\text{primaries} \\ p \in \mathbb{R}}} \chi_{h,p}^{(1)}(\beta_L, \beta_R) + \sum_{\substack{\text{primaries} \\ p \in i\mathbb{R}}} \chi_{h,p}^{(2)}(\beta_L, \beta_R),
\end{aligned}
\tag{47}
$$

where the sum runs over all the primary states labeled by $(h, p)$. Note that the nomenclature "unitary" versus "non-unitary" in this context is coming from the canonical ensemble analysis, and we will use it to make the parallels more clear.

### 3.2 Near-extremal limit

Since the partition function in the quadratic ensemble resembles that of a CFT$_2$, we will consider the same near-extremal limit (see App. B). In particular, the regime of interest is

$$\beta_L \sim c \gg 1, \quad \beta_R \sim c^{-\alpha} \ll 1, \tag{48}$$

where $\alpha > 0$. As we did in the canonical ensemble, we need to make sure we have the necessary conditions to argue that we can project the partition function onto the vacuum character in the limit (48).

We start by looking at the **unitary** case, where we would have

$$\frac{Z(\beta_L, \beta_R)_{\text{u}}}{\chi_{\text{vac}}^{(3)}\left(\frac{4\pi^2}{\beta_L}, \frac{4\pi^2}{\beta_R}\right)} = 1 + \frac{1}{1 - e^{-\frac{4\pi^2}{\beta_L}}} \sum_{\text{primaries}} e^{-\frac{4\pi^2}{\beta_L}(h - p^2/\kappa + 1/12)} e^{\frac{4\pi^2}{\beta_R}(p^2 - p_{\text{vac}}^2)/\kappa}, \tag{49}$$

where we used (46). When $\langle \mathcal{P}_0 \rangle$ is unbounded from below, which is natural to assume from (42), then the sum over $p$ diverges. Therefore, as in the canonical ensemble, the near-extremal regime (48) does not project onto the vacuum.

For non-unitary WCFTs in the quadratic ensemble, the expression one would obtain is similar to (49). The important difference is that from (42) it is natural to assume that the spectrum is bounded from below. Hence, to project onto the vacuum we assume that the spectrum of $\langle \mathcal{P}_0 \rangle$ is bounded from below and that it reaches its minimum for the vacuum.[9] Then, $\beta_R \ll 1$ projects onto the vacuum, and we can write

$$\frac{Z(\beta_L, \beta_R)_{\text{u}}}{\chi_{\text{vac}}^{(2)}\left(\frac{4\pi^2}{\beta_L}, \frac{4\pi^2}{\beta_R}\right)} = 1 + \mathcal{O}\left(e^{-\frac{4\pi^2}{\beta_R}\langle \mathcal{P}_0 \rangle_{\text{gap}}}\right), \tag{50}$$

where $\langle \mathcal{P}_0 \rangle_{\text{gap}} = \min.(\langle \mathcal{P}_0 \rangle - \langle \mathcal{P}_0 \rangle_{\text{vac}})$ and

$$\chi_{\text{vac}}^{(2)}\left(\frac{4\pi^2}{\beta_L}, \frac{4\pi^2}{\beta_R}\right) \approx 2\pi \left(\frac{2\pi}{\beta_L}\right)^2 \exp\left(\frac{\beta_L}{12} - \frac{4\pi^2}{\beta_L}\left(\langle \mathcal{L}_0 \rangle_{\text{vac}} + \frac{1}{12}\right) - \frac{4\pi^2}{\beta_R}\langle \mathcal{P}_0 \rangle_{\text{vac}}\right). \tag{51}$$

Note that we have ignored the corrections $\mathcal{O}(e^{-\beta_L/2}, 1/\beta_L^2)$, which are suppressed by the limit. Thus, the near-extremal partition function in the quadratic ensemble is

$$Z(\beta_L, \beta_R)_{\text{u}} \approx 2\pi \left(\frac{2\pi}{\beta_L}\right)^2 \exp\left(\frac{\beta_L}{12} + \frac{4\pi^2}{\beta_L}\frac{c}{24} - \frac{4\pi^2}{\beta_R}\langle \mathcal{P}_0 \rangle_{\text{vac}}\right), \tag{52}$$

where we used (43) and $c \gg 1$. Here we are taking $p_{\text{vac}}$ to be imaginary, and hence $\langle \mathcal{P}_0 \rangle_{\text{vac}} < 0$. Notice that this expression has some resemblance to a warped-Schwarzian theory, but also significant differences. The polynomial scaling in $\beta_L$ does count the correct number of zero modes as explained below (25). However, this expression is missing an anomalous contribution from the $S$-transform, which correlates with the first term (1). At best, this result seems to correspond to a warped-Schwarzian where $K = 0$, i.e., the $\hat{u}(1)$-level plays no role.

The exponential terms in (52) have a similar form as compared to the near-extremal partition function of the CFT$_2$ (B.15). Thus, we can again go to fixed $(\beta, J)$ and fixed $(\theta, E)$ ensembles and expect similar results. In analogy to the CFT$_2$, we are defining $J \equiv \langle \mathcal{P}_0 \rangle - \langle \mathcal{L}_0 \rangle$ and $E \equiv \langle \mathcal{P}_0 \rangle + \langle \mathcal{L}_0 \rangle$.

---

[9]Note that this is in line with the assumption made in the canonical ensemble, namely (20).

**Fixed $(\beta, J)$ ensemble.** This analysis resembles step by step the CFT$_2$, which is reviewed in App. B. Here we just write the key equations. The ensemble is defined by

$$Z_J(\beta) = \int_{-\pi}^{\pi} \frac{\mathrm{d}\theta}{2\pi} e^{i\theta J} Z(\beta_L, \beta_R)_{\slashed{\nu}}. \tag{53}$$

Here we are using $\beta_L = \beta - i\theta$ and $\beta_R = \beta + i\theta$. Using (52), the location of the saddle point is at

$$i\theta_* = -\beta + 2\pi \sqrt{\frac{-\langle \mathcal{P}_0 \rangle_{\mathrm{vac}}}{(J - 1/12)}}. \tag{54}$$

Note that the saddle is real since $\langle \mathcal{P}_0 \rangle_{\mathrm{vac}} < 0$. To be within the near-extremal regime (48), we furthermore scale $J \sim c^{2\alpha} \gg 1$. In the near-extremal limit, we then have

$$Z_J(\beta) \approx 144\pi \left( -\frac{4\langle \mathcal{P}_0 \rangle_{\mathrm{vac}}}{c^8 J^3} \right)^{\frac{1}{4}} \exp\left( 2\pi \sqrt{-\langle \mathcal{P}_0 \rangle_{\mathrm{vac}} J} - \beta J \right) \left( \frac{c\pi}{12\beta} \right)^2 \exp\left( \frac{\pi^2}{12} \frac{c}{\beta} \right). \tag{55}$$

It is interesting to note that one gets a prefactor of $\beta^{-2}$ as compared to the fixed $(\beta, J)$ ensemble in a CFT or the canonical ensemble of a WCFT, where one gets $\beta^{-3/2}$. This is in contrast to the canonical ensemble (36) due to the absence of the anomaly in the modular transformation (45). One could argue that the $\beta$ dependence in (55) is due to a warped-Schwarzian theory with $K = 0$, but the zero mode of the internal symmetry still contributes to the path integral.

**Fixed $(\theta, E)$ ensemble.** Like in appendix B, one could formally go to a fixed $(\theta, E)$ ensemble as follows

$$Z_E(\theta) = \int_0^{\infty} \mathrm{d}\beta\, e^{\beta E} Z(\beta_L, \beta_R)_{\slashed{\nu}}. \tag{56}$$

However, just like in (B.26), it turns out that this ensemble is ill-defined.

## 4 Discussion

In this final section, we will summarize our main findings with emphasis on comparing and contrasting the different features we found.

**Comparing CFT$_2$ and WCFT.** It is instructive to first compare the extremal limits in a WCFT (24) in the canonical ensemble versus a CFT$_2$ (B.9). Not surprisingly, in both cases we are required to take a large-$c$ limit. The role of the potentials is slightly different since scaling symmetry acts differently on the variables. For our conventions, $\vartheta$ plays the role of $\beta_L$, and $\Omega$ plays the role of $\beta_R$.

The near-extremal limit imposed some restrictions on the spectrum for each case. For a CFT$_2$ these are mild: compactness, unitarity, and no additional symmetries.[10] The WCFT also required compactness and no additional symmetries. However, we also needed the theory to be non-unitary. And the vacuum state had to furthermore comply with the conditions described around (20).

We can make one bold statement based on this last finding. If we assume that the universality of the near-extremal limit, and therefore the existence of Schwarzian-like sector, is a necessary condition for the holographic properties of near-extremal black holes, our results would imply that a unitary WCFT cannot be dual to a semi-classical theory of gravity. More

---

[10]It is possible to relax this last assumption and include currents. They add more zero modes to the 1D theory, and hence dress the Schwarzian theory but the limit still functions.

precisely, it cannot be dual to a semi-classical theory that admits as solutions geometries with an $AdS_2$ factor. This would explain why until now all holographic WCFTs that have been identified are non-unitary.

Finally, there are also some interesting properties to contrast regarding the near-extremal partition function.

1. In the WCFT, it was consistent to convert our answers to the $(\beta, J)$ and $(\vartheta, E)$ ensemble. In the $CFT_2$, only the $(\beta, J)$ ensemble is consistent.

2. The $(\beta, J)$ ensemble for WCFT in the canonical ensemble had two distinct features. First, we found the curious restriction that $J$ has to be large and negative. It also had the curious feature that the heat capacity depended on $J$. This dependence can obviously be reabsorbed in $\beta$, but we will see in [16] that the gravitational dual also exhibits this dependence.

3. In the canonical ensemble, the near-extremal partition function of a WCFT would give an answer very similar to the near-extremal limit of a $CFT_2$ with a global abelian symmetry. The characters and modular properties are the same in both cases; see, for example, [23]. However, for a CFT with a global $U(1)$ symmetry, we would have in (29) that $p_{vac} = 0$ and $S_0$ would be coming from the anti-holomorphic sector.

**Drawbacks of the quadratic ensemble.** The expressions we found in the quadratic ensemble are somewhat peculiar. In contrast to (25) for the canonical ensemble, equation (52) does not fit the warped-Schwarzian theory, because the partition function is modular invariant. It also does not resemble the near-extremal limit of a $CFT_2$, nor a $CFT_2$ with a global abelian symmetry. From a holographic perspective, we expect this difference to be due to boundary conditions for gauge fields in $AdS_2$, a point we will discuss in [16].

# Acknowledgements

**Funding information** AA is a Research Fellow of the Fonds de la Recherche Scientifique F.R.S.-FNRS (Belgium). AA is partially supported by IISN – Belgium (convention 4.4503.15) and by the Delta ITP consortium, a program of the NWO that is funded by the Dutch Ministry of Education, Culture and Science (OCW). The work of AC has been partially supported by STFC consolidated grant ST/T000694/1. B.M. is supported in part by the Simons Foundation Grant No. 385602 and the Natural Sciences and Engineering Research Council of Canada (NSERC), funding reference number SAPIN/00047-2020. SD is a Senior Research Associate of the Fonds de la Recherche Scientifique F.R.S.-FNRS (Belgium). SD was supported in part by IISN – Belgium (convention 4.4503.15) and benefited from the support of the Solvay Family. SD acknowledges support of the Fonds de la Recherche Scientifique F.R.S.-FNRS (Belgium) through the CDR project C 60/5 - CDR/OL "Horizon holography: black holes and field theories" (2020-2022), and the PDR/OL C62/5 project "Black hole horizons: away from conformality" (2022-2025).

# A   Cardy formula for WCFTs

In this appendix, we re-derive the Cardy formula for a WCFT both in the canonical as well as the quadratic ensemble. We highlight the main differences between the Cardy limit and the near-extremal limit.

**Canonical ensemble.** For the canonical ensemble, the torus partition function for a **unitary** (13) and **non-unitary** (14) WCFT are

$$Z(\tau, z)_{\mathrm{u}} = \sum_{\substack{\text{primaries} \\ +\text{vacuum}}} \chi_{h,p}^{(3)}(\tau, z), \tag{A.1}$$

and

$$Z(\tau, z)_{\cancel{\mathrm{u}}} = \sum_{\substack{\text{primaries} \\ p \in \mathbb{R}}} \chi_{h,p}^{(1)}(\tau, z) + \sum_{\substack{\text{primaries} \\ p \in i\mathbb{R}}} \chi_{h,p}^{(2)}(\tau, z), \tag{A.2}$$

respectively. The Cardy behaviour is defined in the regime of small angular potential $\vartheta \ll 1$. To project onto the vacuum, we also need to assume that the spectrum of $L_0$ is bounded from below. Additionally we assume that the WCFT has no state with $h = h_{\mathrm{vac}}$ other than the vacuum itself. This is the counterpart of the existence of a twist gap in the CFT (see App. B). Note that the Cardy limit does not impose any conditions on $p_{\mathrm{vac}}$. It can either be real or imaginary.[11] In the Cardy-limit, $\vartheta \ll 1$ the behaviour of the modular transformed **unitary** (A.1) and **non-unitary** (A.2) torus partition functions depends in both cases on the vacuum character

$$\chi_{h,p}^{\mathrm{vac}}\left(-\frac{1}{\tau}, \frac{z}{\tau}\right) \approx e^{-2\pi i \frac{1}{\tau}(h_{\mathrm{vac}} - \frac{c}{24})} e^{2\pi \frac{z}{\tau} i p_{\mathrm{vac}}}. \tag{A.3}$$

The vacuum character for the non-unitary case can either be $\chi_{h,p}^{(1)}$ (10) or $\chi_{h,p}^{(2)}$ (11) since to leading order the $-1/\tau \to i\infty$ expansions of $\eta(-2/\tau)$ and $\eta(-1/\tau)^2$ agree with each other. We obtain the torus partition function

$$Z(\tau, z) \approx e^{-i\pi\kappa \frac{z^2}{2\tau}} \chi_{h,p}^{\mathrm{vac}}\left(-\frac{1}{\tau}, \frac{z}{\tau}\right), \tag{A.4}$$

where we omitted a subscript to indicate that (A.4) is valid in both the unitary and non-unitary case. The Cardy formula for the entropy now follows from (A.4)

$$S_{\mathrm{Cardy}} = (1 - z\partial_z - \tau\partial_\tau)Z(\tau, z) = 2\pi i\left(\frac{z}{\tau}\langle P_0\rangle_{\mathrm{vac}} - \frac{2}{\tau}\langle L_0\rangle_{\mathrm{vac}}\right), \tag{A.5}$$

where we used that on the cylinder the eigenvalues of $\langle L_0\rangle_{\mathrm{vac}}$ and $\langle P_0\rangle_{\mathrm{vac}}$ are

$$\langle L_0\rangle_{\mathrm{vac}} = h_{\mathrm{vac}} - \frac{c}{24}, \qquad \langle P_0\rangle_{\mathrm{vac}} = p_{\mathrm{vac}}. \tag{A.6}$$

Using (5), the expression (A.5) agrees with the well-known warped Cardy formula [3].

**Quadratic ensemble.** For the quadratic ensemble, we have the characters (46). The torus partition function is (47), which we schematically write as

$$Z(\beta_L, \beta_R) = \sum_{\substack{\text{primaries} \\ +\text{vacuum}}} \chi_{h,p}(\beta_L, \beta_R). \tag{A.7}$$

We do not distinguish between unitary and non-unitary theories since they lead to the same final results (A.8) and (A.9). The Cardy regime is $\beta_L \ll 1$. The only conditions necessary, otherwise, are that the vacuum is the only state which satisfies $h_{\mathrm{vac}} = p_{\mathrm{vac}}^2/\kappa$ and we assume a non-trivial positive gap $\langle \mathcal{L}_0\rangle_{\mathrm{gap}} > 0$ between $\langle \mathcal{L}_0\rangle_{\mathrm{vac}}$ and the other non-vacuum primaries.

---

[11]For the entropy to be real, we need either $p_{\mathrm{vac}} = 0$ or $p_{\mathrm{vac}} \in i\mathbb{R}$.

No condition on $\beta_R$ or the spectrum of $\mathcal{P}_0$ is necessary to obtain the quadratic ensemble Cardy entropy. Thus, when $\beta_L \ll 1$, we find

$$Z(\beta_L, \beta_R) = e^{-\frac{4\pi^2}{\beta_L}\langle \mathcal{L}_0 \rangle_{\text{vac}} - \frac{4\pi^2}{\beta_R}\langle \mathcal{P}_0 \rangle_{\text{vac}}}, \tag{A.8}$$

where the eigenvalues $\langle \mathcal{L}_0 \rangle_{\text{vac}}$ and $\langle \mathcal{P}_0 \rangle_{\text{vac}}$ are defined in (43). We then obtain the warped Cardy formula for the quadratic ensemble as found in [3]

$$S_{\text{Cardy}} = -\left( \frac{8\pi^2}{\beta_L}\langle \mathcal{L}_0 \rangle_{\text{vac}} + \frac{8\pi^2}{\beta_R}\langle \mathcal{P}_0 \rangle_{\text{vac}} \right). \tag{A.9}$$

This matches the warped Cardy formula in the quadratic ensemble found in [3]. Note that $\langle \mathcal{L}_0 \rangle_{\text{vac}} = -c/24$ and $\langle \mathcal{P}_0 \rangle_{\text{vac}} \leq 0$ for a unitary theory or a non-unitary theory with $p_{\text{vac}} \in i\mathbb{R}$. So, the entropy is positive in these cases but when $p_{\text{vac}} \in \mathbb{R}$ for a non-unitary theory such that $\langle \mathcal{P}_0 \rangle_{\text{vac}} > 0$, it should be interpreted as an index.

# B  Near-extremal limits in CFT$_2$

In this appendix, we provide a review of the near-extremal limit of partition functions of two-dimensional CFTs; this was done originally in [5]. The analysis in this appendix should be contrasted with the WCFT cases, which we discuss in the main text.

We consider a Euclidean unitary two-dimensional conformal field theory with central charge $c$. We will place the theory on a torus parametrised by an angular coordinate $\varphi$ and the Euclidean time coordinate $t_E$. The metric on the torus and the identifications are

$$ds^2 = dt_E^2 + d\varphi^2, \quad (t_E, \varphi) \sim (t_E + \beta, \varphi + \theta) \sim (t_E, \varphi + 2\pi), \tag{B.1}$$

where $\theta$ is the twist angle and $\beta = T^{-1}$ is the inverse temperature. The complex structure of the torus is therefore given by

$$\tau = (\theta + i\beta)/2\pi, \qquad \bar{\tau} = (\theta - i\beta)/2\pi. \tag{B.2}$$

It will be also useful to introduce right and left moving potentials; they are defined as

$$\beta = \frac{1}{2}(\beta_R + \beta_L), \qquad \theta = \frac{1}{2i}(\beta_R - \beta_L), \tag{B.3}$$

and they are related to the complex structure as

$$\tau = \frac{i\beta_L}{2\pi}, \qquad \bar{\tau} = -\frac{i\beta_R}{2\pi}. \tag{B.4}$$

Notice that when $\theta$ is purely imaginary and $\beta$ is real, then $\beta_{R/L} \in \mathbb{R}$.

The partition function on the torus is given by

$$Z(\beta_L, \beta_R) = \text{Tr}\left( e^{-\beta_L\left(L_0 - \frac{c}{24}\right) - \beta_R\left(\bar{L}_0 - \frac{c}{24}\right)} \right), \tag{B.5}$$

where $L_n$, with $n \in \mathbb{Z}$, are the Virasoro generators on the plane. For the zero modes, we have that $H = L_0 + \bar{L}_0 - \frac{c}{12}$ and $J = \bar{L}_0 - L_0$, with $H$ and $J$ the Hamiltonian and angular momentum, respectively. The eigenvalues of $L_0$ and $\bar{L}_0$ will be denoted as $h$ and $\bar{h}$. In the subsequent derivations we assume that the CFT$_2$ is unitary (spectrum bounded from below), the stress tensor is the only current (we only have Virasoro symmetry), and the theory is compact (spectrum is discrete). The last two assumptions imply the existence of a twist gap,

i.e. a lowest positive conformal dimension $\bar{h}_{\text{gap}} > 0$ between the vacuum and any non vacuum primary. These three assumptions make it then convenient to decompose the partition function as follows,

$$Z(\beta_L, \beta_R) = \chi_{\mathbb{I}}(\tau) \bar{\chi}_{\mathbb{I}}(\bar{\tau}) + \sum_{\substack{\text{primaries} \\ h, \bar{h} > 0}} \chi_h(\tau) \bar{\chi}_{\bar{h}}(\bar{\tau}) . \tag{B.6}$$

Here $\chi_h$ is the Virasoro character of a primary state of weight $h$, with $h \neq 0$, and $\chi_{\mathbb{I}}$ are the Virasoro characters of the vacuum state. Their explicit expression is

$$\chi_h(\tau) = \frac{e^{2\pi i \tau \left(h - \frac{c-1}{24}\right)}}{\eta(\tau)} , \qquad \chi_{\mathbb{I}}(\tau) = (1 - e^{2\pi i \tau}) \frac{e^{-2\pi i \tau \frac{c-1}{24}}}{\eta(\tau)} , \tag{B.7}$$

where $\eta(\tau)$ is the Dedekind eta function, and the expressions are analogous for $\bar{\chi}_{\bar{h}}(\bar{\tau})$. For the vacuum character $\chi_{\mathbb{I}}$, with $h_{\text{vac}} = 0$, we have removed a null state, relative to $\chi_h$.

One interesting result in [5] was to demonstrate that there is a universal behaviour of the torus partition function in the so-called "near-extremal" limit. The definition of this limit in the CFT is as follows. We will consider a class of theories where it is meaningful to take a large central charge limit, i.e.,

$$c \gg 1 , \tag{B.8}$$

and within this regime we will scale the left and right moving potential as

$$\beta_L \sim c \gg 1 , \quad \beta_R \sim c^{-\alpha} \ll 1 , \tag{B.9}$$

with $\alpha > 0$. Introducing the parameter $\alpha$ here is a slight generalization of [5] where they have $\alpha = 1$; as we will see the approximations still hold with this modification.[12] Note that the "near-extremal" is capturing a low temperature regime of the $\text{CFT}_2$ at large central charge, i.e., $T \sim 1/c \ll 1$.

To extract the universal behaviour in the near-extremal limit, we use invariance under $S$-transformations, $(\tau, \bar{\tau}) \to (-1/\tau, 1/\bar{\tau})$, of the torus partition function, that is,

$$\begin{aligned} Z(\beta_L, \beta_R) &= Z\left(\frac{4\pi^2}{\beta_L}, \frac{4\pi^2}{\beta_R}\right) \\ &= \chi_{\mathbb{I}}\left(\frac{2\pi i}{\beta_L}\right) \chi_{\mathbb{I}}\left(\frac{2\pi i}{\beta_R}\right) + \sum_{\text{primaries}} \chi_h\left(\frac{2\pi i}{\beta_L}\right) \chi_{\bar{h}}\left(\frac{2\pi i}{\beta_R}\right) . \end{aligned} \tag{B.10}$$

Implementing the near-extremal limit (B.9), one gets that the partition function (B.10) behaves as

$$\frac{Z(\beta_L, \beta_R)}{\chi_{\mathbb{I}}\left(\frac{2\pi i}{\beta_L}\right) \chi_{\mathbb{I}}\left(\frac{2\pi i}{\beta_R}\right)} = 1 + \mathcal{O}\left(e^{-\frac{4\pi^2}{\beta_R} \bar{h}_{\text{gap}}}\right) . \tag{B.11}$$

This is the key statement regarding universality: the partition function is dominated by the vacuum characters, and corrections are exponentially suppressed. It is important to stress here that the scaling of $\beta_R$ is key in the validity of the approximation. The ratio $\chi_h / \chi_{\mathbb{I}}$ for the left movers is diverging polynomially in $c$ due to the null state in (B.7); this effect gets suppressed since we also set $\beta_R \sim c^{-\alpha}$ and there is a twist gap with $\bar{h}_{\text{gap}} > 0$.

We can further cast (B.11) by taking the near-extremal limit on the vacuum characters. By using two standard properties of the Dedekind eta function,

$$\begin{aligned} \eta(-1/\tau) &= \sqrt{-i\tau}\, \eta(\tau) , \\ \eta(\tau) &\underset{\tau \to i\infty}{\approx} e^{\pi i \tau / 12} , \end{aligned} \tag{B.12}$$

---

[12]From a perspective of $\text{AdS}_3/\text{CFT}_2$ it natural to take $\beta_R \sim c^{-1}$. Here we are making a slightly more general analysis.

one finds that the right moving character projects onto the vacuum

$$\chi_{\mathbb{I}}\left(\frac{2\pi i}{\beta_R}\right) \approx e^{\frac{c}{24}\frac{4\pi^2}{\beta_R}} \,, \tag{B.13}$$

and for the left moving sector we find

$$\chi_{\mathbb{I}}\left(\frac{2\pi i}{\beta_L}\right) \approx 2\pi\left(\frac{2\pi}{\beta_L}\right)^{3/2} e^{\frac{\beta_L}{24}+\frac{c}{24}\frac{4\pi^2}{\beta_L}} \,, \tag{B.14}$$

which includes a polynomial behaviour and exponential behaviour as $c \gg 1$. Combining these two results, we obtain

$$Z(\beta_L, \beta_R) \approx 2\pi\left(\frac{2\pi}{\beta_L}\right)^{3/2} e^{\frac{\beta_L}{24}+\frac{c}{24}\frac{4\pi^2}{\beta_L}+\frac{c}{24}\frac{4\pi^2}{\beta_R}} \,, \tag{B.15}$$

which holds in the near-extremal limit (B.8)-(B.9) and for $\text{CFT}_2$ that comply with the assumptions described between (B.5)-(B.6).

As a final portion of the analysis, it is very instructive to transform (B.15) to mixed ensembles, where one of the charges is fixed. In the following, we will describe the basic thermodynamic variables in the fixed $(\beta, J)$ and fixed $(\theta, M)$ ensemble. We will discuss similarities and differences between these two ensembles, and it is also instructive to contrast these ensembles with the analogous results for a WCFT. For this purpose it is useful to record (B.15) as a function of $(\beta, \theta)$:

$$Z(\beta, \theta) \approx 2\pi\left(\frac{2\pi}{\beta-i\theta}\right)^{3/2} e^{\frac{\beta-i\theta}{24}+\frac{c\pi^2}{3}\frac{\beta}{\beta^2+\theta^2}} \,. \tag{B.16}$$

**Fixed $(\beta, J)$ ensemble.** In the fixed angular momentum ensemble we consider the Fourier transform

$$Z_J(\beta) = \int_{-\pi}^{\pi} \frac{d\theta}{2\pi} e^{i\theta J} Z(\beta, \theta) \,. \tag{B.17}$$

Solving this integral by saddle point approximation in the near-extremal regime, we find that the saddle point $(\theta_*)$ is determined by

$$\frac{\beta\theta_*}{(\beta^2+\theta_*^2)^2} = \frac{3i}{2\pi^2 c}\left(J-\frac{1}{24}\right) \,, \tag{B.18}$$

where we used (B.16). We only need to solve this equation in the near-extremal limit, for which $\beta \sim c$; in this regime we find[13]

$$i\theta_* \approx -\beta + \sqrt{\frac{c\pi^2}{6(J-1/24)}} \,. \tag{B.19}$$

This expression assumes that $\beta$ is much larger than $\sqrt{c/J}$; this can easily be achieved when $J \gg c$, as done in [5], but not necessary for the validity of the saddle point. Still, for the saddle point to be consistent with (B.9), i.e., requiring $\beta_R \sim c^{-\alpha}$, we would need to take $J \sim c^{2\alpha+1}$. For this reason, we will take $J$ to be large in the subsequent expressions. It is also important to note that the saddle point $\theta_*$ is not within the integration range in (B.17). However, we can deform the contour such that it passes (B.19) without crossing any poles.

---

[13]There are four roots to (B.18). Here we are selecting the one that leads to a real and positive value of $Z_J(\beta)$, and the entropy, below.

Finally, after doing the saddle point approximation of (B.17), we find that

$$Z_J(\beta) \approx 12\pi \left(\frac{24}{c^5 J^3}\right)^{1/4} e^{2\pi\sqrt{\frac{cJ}{6}}} e^{-\beta E_0} Z_{\text{schw.}}\left(\frac{12\beta}{c}\right). \tag{B.20}$$

Here we have organized the final answer in terms of three contributions. The first factor is a normalization arising from the saddle point approximation that is independent of $\beta$. This could be interpreted as "extremal entropy," with the leading contribution being

$$S_0 \approx 2\pi\sqrt{\frac{cJ}{6}}, \tag{B.21}$$

plus logarithmic corrections in $c$ and $J$. However, the system does not have a gap in this regime, and therefore this is not the ground state entropy. The second factor is an exponential term, which can be recognized as the "extremal" energy, where at leading order in $J$ we have

$$E_0 = J. \tag{B.22}$$

And the final contribution to (B.20) is the so-called Schwarzian partition function

$$Z_{\text{schw.}}(\tilde{\beta}) \equiv \left(\frac{\pi}{\tilde{\beta}}\right)^{3/2} e^{\frac{\pi^2}{\tilde{\beta}}}. \tag{B.23}$$

Notice that it enters only as a function of $\tilde{\beta}$, which in this case is $\tilde{\beta} = \frac{12\beta}{c}$.

**Fixed $(\theta, E)$ ensemble.** Albeit not a good choice, let us naively consider an ensemble where $E$ and $\theta$ are fixed. Formally, it is defined as

$$Z_E(\theta) = \int_0^\infty d\beta \, e^{\beta E} Z(\beta, \theta). \tag{B.24}$$

The procedure is very similar as the fixed $(\beta, J)$ ensemble, so we will just report on the main findings here. The saddle point of (B.24) is located at

$$\beta_* \approx -i\theta + 2\pi\sqrt{\frac{c}{24E+1}}, \tag{B.25}$$

where this solution approximates $\text{Im}(\theta) \sim c \gg 1$. Requiring consistency with the near extremal limit (B.9) implies that $E \sim c^{2\alpha+1}$. The saddle point integral then gives

$$Z_E(\theta) \approx 24\pi^2 i \left(\frac{24}{c^5 E^3}\right)^{1/4} e^{2\pi\sqrt{\frac{cE}{6}}} e^{-\text{Im}(\theta)J_0} Z_{\text{schw.}}\left(\frac{12}{c}\text{Im}(\theta)\right), \tag{B.26}$$

where $J_0 = -E$, and $Z_{\text{schw.}}(\tilde{\beta})$ is defined in (B.23) with an effective temperature of $\tilde{\beta} = \frac{12}{c}\text{Im}(\theta)$.

The fact that the ground state energy is negative and the appearance of an $i$ in the partition function indicate that this is not a physical choice for the ensemble. This could be removed if one takes $E \to -E$ which is clearly undesirable physically, and it also affects (B.25). The near-extremal limit does not treat $\beta$ and $\theta$ on the same footing.



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
