# Peer review of "Near-Extremal Limits of Warped CFTs"

_SciPost Physics, doi:SciPost Phys. 15, 056 (2023)_

## Round 2 · Referee Report · Anonymous (Referee 1) · 2023-6-1

Strengths

1-The paper carefully defines the near-extremal limit of torus partition function of WCFTs under which a universal structure appears, and thus fills in an important gap in the existing literature.
2-They find that such a limit can be only accessed in non-unitary WCFTs, but not in the unitary ones.
3-The differences between the so-called canonical ensemble and quadratic ensembles are carefully analyzed.

Report

The paper studies warped conformal field theories (WCFTs), which are two dimensional quantum systems featuring a chiral Virasoro-Kac-Moody algebra. WCFTs have been conjectured to be holographically dual to gravitational systems with black hole solutions. Meanwhile, they are interesting in their own right. It has been shown in the previous literature that the near-extremal limit of black holes is universal and has a sector that can be described by a Schwartzian action. The paper carefully defines the near-extremal limit of torus partition function of WCFT under which the vacuum state dominates and the warped version of Schwartzian effective action appears. They find that such a limit can only be accessed in non-unitary WCFTs, but not in the unitary ones. They also compare the differences between the so-called canonical ensemble and quadratic ensembles. The careful analysis carried out in the paper is useful for further sharpening the definition of WCFTs and also for setting up more details in the holographic duality. The paper is very well-written. I recommend the paper to be published on Scipost.

Requested changes

1-"...unitary WCFTs do no have ..."in the sentence above "Non-Unitary WCFTs" should be "...unitary WCFTs do not have ...".

---

## Round 2 · Referee Report · Anonymous (Referee 2) · 2023-6-2

Strengths

1-Well written and clear
2-Attacks and solves a well defined problem
3-Solves a previous outstanding puzzle

Weaknesses

1-The paper presents only a small technical result
2-The physical consequences of this result are not fully explored or applied.

Report

This paper presents a clear result to an outstanding puzzle. It is well written and self-contained. While of technical nature, it will be of use to researchers in the field. I recommend publication.

Requested changes

None.

---

## Editorial Decision

published